# Recent Progress in Gene-Targeting Therapies for Spinal Muscular Atrophy: Promises and Challenges

**DOI:** 10.3390/genes15080999

**Published:** 2024-07-30

**Authors:** Umme Sabrina Haque, Toshifumi Yokota

**Affiliations:** 1Department of Neuroscience, Faculty of Medicine and Dentistry, University of Alberta, Edmonton, AB T6G 2H7, Canada; ummesabr@ualberta.ca; 2Department of Medical Genetics, Faculty of Medicine and Dentistry, University of Alberta, Edmonton, AB T6G 2H7, Canada; 3The Friends of Garrett Cumming Research & Muscular Dystrophy Canada HM Toupin Neurological Science Research, Edmonton, AB T6G 2H7, Canada

**Keywords:** spinal muscular atrophy (SMA), survival of motor neuron 1 (*SMN1*), *SMN2*, SMN protein, antisense oligonucleotide (ASO), nusinersen, gene therapy, onasemnogene, risdiplam, small molecule, combination therapy

## Abstract

Spinal muscular atrophy (SMA) is a severe genetic disorder characterized by the loss of motor neurons, leading to progressive muscle weakness, loss of mobility, and respiratory complications. In its most severe forms, SMA can result in death within the first two years of life if untreated. The condition arises from mutations in the *SMN1* (survival of motor neuron 1) gene, causing a deficiency in the survival motor neuron (SMN) protein. Humans possess a near-identical gene, *SMN2*, which modifies disease severity and is a primary target for therapies. Recent therapeutic advancements include antisense oligonucleotides (ASOs), small molecules targeting SMN2, and virus-mediated gene replacement therapy delivering a functional copy of SMN1. Additionally, recognizing SMA’s broader phenotype involving multiple organs has led to the development of SMN-independent therapies. Evidence now indicates that SMA affects multiple organ systems, suggesting the need for SMN-independent treatments along with SMN-targeting therapies. No single therapy can cure SMA; thus, combination therapies may be essential for comprehensive treatment. This review addresses the SMA etiology, the role of SMN, and provides an overview of the rapidly evolving therapeutic landscape, highlighting current achievements and future directions.

## 1. Introduction

Spinal muscular atrophy (SMA) is one of the most prevalent autosomal recessive genetic disorders, affecting approximately 1 in 10,000 births [1]. SMA is mainly characterized by the degeneration of α motor neurons located in the anterior horn of the spinal cord and motor nuclei of the lower brainstem, resulting in muscle wasting, weakness, hypotonia, and difficulties with feeding and respiration [2,3,4,5].

The underlying cause of classic spinal muscular atrophy (SMA) is typically the homozygous absence or, less commonly, smaller mutations within the *SMN1* gene located on chromosome 5. Additionally, due to intrachromosomal duplication, humans have another paralogous gene, *SMN2* [6]. However, SMN2 cannot compensate for the loss of SMN1 due to a single-nucleotide polymorphism in exon 7 (C-o-T transition), which disrupts an exonic splicing enhancer (ESE) or creates an exonic splicing silencer (ESS). The ESE and ESS are cis-acting exonic sequences that influence the use of flanking splice sites, leading to the exclusion of exon 7 in approximately 90% of SMN2 transcripts. This results in the production of a truncated and unstable protein, SMNΔ7, which is rapidly degraded [7]. The genetic alterations in SMN1 result in decreased expression of the survival motor neuron protein (SMN). The SMN protein is expressed ubiquitously in almost every cell, both in the nucleus and cytoplasm, and it plays a crucial role in various cellular mechanisms. These include the assembly of spliceosomal machinery, endocytosis, protein translation, and maintenance of cellular homeostasis. Due to its diverse functions and widespread expression, the loss of SMN can result in systemic pathology extending beyond the motor neuron [8,9].

Several therapeutic strategies, such as nusinersen, onasemnogene abeparvovec, and risdiplam, have gained regulatory approval by the FDA and EMA for the treatment of SMA. These approaches aim to enhance SMN production either by modifying SMN2 splicing or by replacing the defective *SMN1* gene [10]. Additionally, various other strategies have been explored, including elevating SMN transcript levels, stabilizing SMN protein, implementing neuroprotective measures, employing muscle activators, and more. While these therapies do not directly address the primary deficiency of SMN, they could be used in conjunction with treatments that boost SMN production to offer additional benefits to patients [11].

This review aims to comprehensively discuss the molecular characteristics, pathophysiology, and currently approved treatments for SMA. Furthermore, we will explore alternative approaches focused on increasing SMN levels.

### 1.1. Genetic Background of SMA and Diverse Role of SMN

SMA is caused by the absence or mutations in the survival of motor neuron gene 1 (*SMN1*), which was originally cloned and characterized by Melki and colleagues [2]. SMN1, also referred to as SMNT (with T standing for telomere), spans 20 kb and is located in the telomeric region of a 500 kb inverted duplication on chromosome 5q13. This genomic locus is characterized by a high number of repetitions, resulting in instability and frequent genetic rearrangements in most SMA patients. Unlike other species, such as mice, which have only one copy of the *SMN* gene, humans have a variable number of centromeric *SMN2* genes [12]. Notably, SMN2 appears to be unique to humans, as chimpanzees have multiple SMN1 copies but no SMN2. SMN2, also referred to as SMNC (with C indicating centromere), shares over 99% nucleotide identity with SMN1. Each SMN gene comprises nine exons, including exons 1, 2a, 2b, 3, 4, 5, 6, 7, and 8 (which encode the 3′ untranslated region (UTR)) responsible for encoding the survival of motor neuron (SMN) protein. However, SMN1 and SMN2 differ by eight nucleotides, five of which are intronic and three of which occur in the last three exons [13]. Among these differences, a single functional coding variant, c.840C>T in exon 7 of SMN2, disrupts an exonic splicing enhancer and simultaneously creates an exonic splicing silencer [14,15] (Figure 1). Normally, SMN1 produces full-length mRNA, which translates into functional SMN protein (294 amino acids, 38 kDa). The *SMN1* gene is essential in diverse organisms, in which null mutations are lethal during early development [16]. In contrast to SMN1, alternative splicing due to the presence of an exonic splicing silencer results in the exclusion of exon 7 in SMN2. This leads to a shortened mRNA that encodes a truncated and unstable SMN protein (SMNΔ7; 282 amino acids, 30.5 kDa), which is rapidly degraded. However, a minority of SMN2 pre-mRNA transcripts retain exon 7 after splicing, leading to the production of approximately 10% of functional protein from SMN2, thereby making it a disease-modifying gene [17] (Figure 1).

The SMN protein, translated from the *SMN* gene, is ubiquitously expressed in all cells and tissues, with particularly high levels in the nervous system, especially the spinal cord [18]. During gestational and neonatal stages, SMN protein expression is high beyond the neuromuscular system and declines with age, but motor neurons of the spinal cord maintain high SMN levels throughout life [19,20]. SMN forms a complex with Gemins2–8 in the cytoplasm and in nuclear bodies called gems, similar to Cajal bodies [21,22]. In addition to gemins, other proteins also interact with SMN. These include Sm, Sm-like proteins, RNA helicase A, fibrillarin, GAR1, ribonucleoprotein (RNP)-heterogenous nuclear RNP U (hnRNP U), hnRNP Q, hnRNP R, and p80-coilin, the marker for Cajal bodies [23,24]. The primary role of SMN is in the biogenesis and maintenance of spliceosomal small nuclear ribonucleoprotein (snRNP) assembly, crucial for pre-mRNA splicing. Additionally, SMN is involved in transcriptional regulation, telomerase regeneration, autophagy, cellular trafficking and homeostasis, signal transduction, DNA repair, and recombination [24].

Structurally, the 294-amino-acid-long SMN protein comprises multiple domains: the N-terminal Gemin2 and nucleic acids binding domain, the central Tudor domain, and the C-terminal proline- and YG box-rich domain. The Tudor domain interacts with the coilin protein, a marker of Cajal bodies. The domain also binds to the C-terminal arginine- and glycine-rich tails of Sm core proteins to facilitate spliceosome assembly. The YG box is a tyrosine/glycine-rich region in the C-terminus of the SMN protein, enabling SMN oligomerization through a glycine zipper structure. Mutations in these domains are linked to SMA, highlighting the importance of SMN’s structural integrity for its function [25].

Reduced expression of SMN protein in SMA leads to impaired α-motor neuron development and degeneration, manifesting as skeletal muscle weakness, the most evident clinical manifestation of SMA [26]. Early stages of SMA involve impaired motor neuron development, affecting the cell body, axons, and myofibers. This results in delayed synaptic input acquisition, immature firing patterns [27,28,29], reduced motor axon growth and myelination [30], and deficient NMJ synapse function [31,32,33]. As the disease progresses, terminal motor axons withdraw from NMJ postsynaptic terminals, proximal motor axons degenerate, and motor neuron cell bodies lose synaptic inputs and undergo cell death [24]. Prolonged denervation of myofibers leads to their replacement by fibro-adipose tissue [34]. Later stages of SMA involve slower neurodegeneration, primarily affecting distal motor components. Key molecular mediators of neurodegeneration in SMA include the p53 pathway, which is activated due to altered splicing of Mdm2 and Mdm4, as well as JNK signaling, ER stress, and DNA damage [24].

Ninety-five percent of affected individuals have the homozygous absence of both exons 7 and 8 or only exon 7 of SMN1, regardless of the disease phenotype. Most of the remaining 5% are typically compound heterozygotes with an SMN1 exon 7 absence and an SMN1 point mutation [35]. In patients with less severe forms of *SMA* (types 2 and 3), gene conversion (where SMN1 exon 7 is replaced by SMN2 exon 7) often occurs instead of genuine deletions of SMN1, which are more common in SMA type 1 [36]. The protein domain encoded by exon 7 (amino acids 280–294) is essential for SMN’s function [7,37]. The SMNΔ7 protein, which lacks this exon, is unstable, rapidly degraded, and deficient in oligomerization [14,38], binding to Sm core proteins [39], and gem formation [40]. SMNΔ7 has a significantly shorter half-life compared to full-length SMN due to degradation via the ubiquitin-proteasome system [41]. However, its stability can be enhanced by coexpressing full-length SMN, which recruits SMNΔ7 into oligomeric complexes [42]. Additionally, exon 7 includes a cytoplasmic targeting signal vital for transporting SMN into neuronal processes [43].

Additional intragenic mutations, including missense, nonsense, splice site mutations, insertions, deletions, and duplications, are particularly common in exons 3 and 6 [44,45]. These missense mutations are of significant interest as they may alter specific properties of the SMN protein without causing a complete loss of function. Most SMA mutations are clustered in the Tudor domain encoded by exon 3 (e.g., W92S, V94G, G95R, A111G, I116F, Y130C, E134K, and Q136E) and in exon 6, within or near the Y/G box (e.g., L260S, S262G, S262I, M263R, M263T, S266P, Y272C, H273R, T274I, G275S, G279C, and G279V). This mutation distribution indicates that both the Tudor domain and the Y/G box, along with their surrounding regions, are critical for SMN function [46].

### 1.2. Clinical Manifestation of SMA

SMA was first described by Guido Werdnig in 1891 in a case involving muscle weakness in two infant brothers, followed by seven additional cases reported by Johan Hoffmann from 1893 to 1900 [47,48]. SMA leads to the progressive loss of α motor neurons in the ventral spinal cord and motor nuclei of the lower brainstem. Clinically, SMA presents as hypotonia, muscle weakness, and atrophy, with severity varying by genotype. The muscle weakness is predominantly proximal, with greater involvement of the lower extremities, and is usually symmetric, accompanied by diffuse areflexia [1]. In severe cases, bulbar and respiratory muscle weakness can occur, although facial and ocular muscles are generally spared [49]. The various phenotypes of spinal muscular atrophy (SMA) were formalized into a classification scheme at the 1991 International Consortium on Spinal Muscular Atrophy, sponsored by the Muscular Dystrophy Association (MDA). This initial classification identified three SMA types based on the highest level of motor function achieved (e.g., sitting or standing) and age of onset. Later modifications further divided the type 3 category by introducing type 4 for adult-onset cases and type 0 for prenatal onset cases resulting in death within weeks [35,47] (Table 1).

## 2. FDA-Approved Gene-Targeting SMN Replacement Therapies

### 2.1. Nusinersen

#### 2.1.1. Outline

Nusinersen (Spinraza^®^; Biogen, Cambridge, MA, USA) is an antisense oligonucleotide (ASO) that was the first treatment approved for SMA. The FDA granted approval in December 2016, followed by EMA approval in May 2017 [54]. Nusinersen works by promoting the inclusion of SMN2 exon 7 via ISS-N1 inhibition, thereby increasing the production of full-length SMN2 mRNA and subsequently full-length SMN protein. For effective delivery to motor neurons in the central nervous system, nusinersen must be administered intrathecally because ASOs do not cross the intact blood-brain barrier. The dosing regimen includes a loading phase of four intrathecal injections over a two-month period (on days 0, 15, 30, and 60) with a dose of 12 milligrams (4–5 milliliters, depending on age). After the loading phase, a maintenance dose is administered every four months to sustain therapeutic levels [55] (Table 2).

#### 2.1.2. Discovery and Mechanism of Action

The discovery of nusinersen is rooted in modifying the splicing pattern of the *SMN2* gene to produce the full-length, functional SMN protein. This process involves targeting splicing regulatory elements to enhance exon 7 inclusion, which is crucial to produce the functional protein. A pivotal moment came with Ravindra N. Singh’s team at the University of Massachusetts Medical School. They identified an intronic splicing silencer (ISS-N1) at the 5′ end of SMN2 intron 7, which strongly inhibited exon 7 inclusion [56]. Deleting or masking the ISS-N1 region can promote the inclusion of exon 7 in most SMN2 transcripts [56]. The use of ASOs to manipulate such regulatory regions has emerged as an effective therapeutic approach. ASOs are single-stranded, DNA-like molecules designed to hybridize to complementary mRNA sequences, thereby modifying gene expression [57]. Singh’s team developed an ASO (7–25) targeting this site, providing a foundation for nusinersen’s development [58]. Unlike nusinersen, which targets the 10–27 region and uses methoxyethyl (MOE) modifications, this ASO utilized 2′-O-methyl (OME) modifications and was less effective in correcting SMN2 splicing in the mild Taiwanese SMA model, also causing an inflammatory response [59]. Other notable achievements include Lim and Hertel’s use of ASOs to bind the 3′-splice site of intron 7, enhancing exon 7 inclusion by promoting intron 6’s 3′SS recognition. Adrian Krainer’s lab at Cold Spring Harbor Laboratory developed peptide nucleic acids to target SMN2 more effectively, achieving efficient exon 7 inclusion in cells. Francesco Muntoni’s lab at Imperial College London employed bifunctional ASOs that recruited splicing activators, significantly boosting SMN protein levels in SMA patient cells. Despite these advancements, early ASOs showed limited success in animal models, primarily due to the lack of systematic studies to determine the optimal ASO binding site and the unclear chemical modifications most suitable for drug development [60]. This underscored the need for further optimization, ultimately leading to the successful development of treatments like nusinersen.

The development of nusinersen involved testing multiple ASOs targeting the ISS-N1 region in mouse models and the SMA fibroblast, with the sequence shifting one base at a time (walking method) to determine the most effective target sequence for exon 7 inclusion by AR Krainer’s lab [61]. The ASO targeting intron 7 position +10-27 was identified as the strongest sequence. This 18-mer 2′-O-methoxyethyl-modified ASO significantly increased SMN protein levels and improved the SMA phenotype in mice, particularly with early treatment [59]. The successful delivery of ASOs to motor neurons in the central nervous system was a significant challenge due to the blood–brain barrier. Williams J et al. overcame this challenge by demonstrating that repeated intracerebroventricular (ICV) administration of a 2′-O-methyl (2′OMe) antisense oligonucleotide increased SMN levels in the central nervous system tissues of SMNΔ7 SMA mice (SMNΔ7+/+; SMN2 +/+; Smn−/−) [58]. Subsequent studies confirmed the efficacy of ICV injections in delivering MOE AONs targeting ISS-N1 to the brain and spinal cord. Preclinical studies in mouse models and non-human primates showed good ASO distribution throughout the spinal cord and dose-dependent effects on SMN expression following a single intrathecal infusion [62,63]. However, subsequent studies underscored the critical role of peripheral SMN restoration using MOE antisense oligonucleotides, which had the strongest effect on both survival and vascular-related clinical signs in a severe mouse model of SMA [64,65].

Nusinersen is a 2′-O-methoxyethyl-modified ASO thatworks by binding to ISS-N1 and inhibiting the binding of other splicing factors, thereby promoting the inclusion of exon 7 into the mRNA. Studies suggest that ISS-N1 interacts with various splicing factors, such as hnRNP A1/A2, which contribute to its inhibitory effect [61]. However, another proposed mechanism is that ISS-N1-targeting ASO alters the secondary structure of pre-mRNA, facilitating the recruitment of U1 snRNP at the 5′ splice site of exon 7. U1 snRNP binds to the 5′ exon-intron junction of pre-mRNA, playing a crucial role in the early stages of splicing [66].

#### 2.1.3. Clinical Trials

The **ENDEAR** trial (NCT02193074) was a 13-month, international, randomized, multicenter, sham-controlled, phase III study designed to evaluate the clinical efficacy and safety of nusinersen in infants diagnosed with spinal muscular atrophy (SMA). These infants had two copies of the *SMN2* gene and exhibited symptoms at or before six months of age. The trial enrolled 121 infants, with two-thirds (81 infants) receiving nusinersen and one-third (41 infants) receiving a sham treatment. The primary endpoints were the proportion of motor milestone responders, assessed by the Hammersmith Infant Neurological Examination Part 2 (HINE2), and event-free survival, evaluated by the time to death or the need for permanent ventilatory support. Secondary endpoints included subgroup analyses of overall survival and event-free survival rates based on the duration of the disease at screening. Results showed that a significantly higher percentage of infants in the nusinersen group achieved motor milestone responses compared to the control group (51% vs. 0%). Additionally, the likelihood of event-free survival was higher in the nusinersen group, though only 8% of treated infants achieved independent sitting. The trial also highlighted the importance of early intervention, showing that infants treated earlier (under three months of age) demonstrated quicker and more pronounced motor improvements. The incidence of adverse events was similar between the nusinersen group (96%) and the control group (98%) [67].

The **CHERISH** trial (NCT02292537) was a phase III study designed to assess the efficacy and safety of nusinersen in children with SMA type II and III who presented symptoms after six months of age. The trial enrolled 126 children, with two-thirds (84 children) receiving nusinersen and one-third (42 children) receiving a sham treatment. The primary endpoint of the study was the least-squares mean change from baseline in the Hammersmith Expanded Functional Motor Scale (HFMSE) score at 15 months post-treatment initiation. The secondary endpoint was the percentage of children who showed a clinically meaningful increase in the HFMSE score, defined as an improvement of at least three points, indicating better performance in at least two motor skills. Interim analysis revealed a significant improvement in motor function for the treated group, with the HFMSE score increasing by a mean of four points after 15 months of treatment. In contrast, the sham group showed a mean decrease of 1.9 points. Final analysis confirmed these findings, with 57% of children in the nusinersen group showing an increase of at least three points in the HFMSE score compared to 26% in the control group (*p* < 0.001). The most frequent adverse events in the nusinersen group were related to the lumbar puncture procedure, such as headaches, back pain, and post-lumbar puncture syndrome. Following the interim analysis, both trials were terminated, and all patients were enrolled in the SHINE (NCT02594124) open-label extension study to continue evaluating long-term safety and efficacy [68].

The **NURTURE** (NCT02386553) study is a phase 2, multisite, open-label, single-arm trial focusing on pre-symptomatic SMA patients likely to develop SMA type I or II. The goal was to evaluate the long-term safety and efficacy of intrathecal nusinersen when initiated early, before the onset of clinical SMA signs. The trial enrolled 25 infants with genetically diagnosed SMA, including 15 with two copies of the *SMN2* gene and 10 with three copies. The primary endpoint was the time to death or the need for respiratory intervention (either invasive or non-invasive for at least 6 h per day for a minimum of 7 days, or tracheostomy). Secondary endpoints included survival rates, motor milestone achievement, and maintenance of weight. Motor milestones were assessed using the World Health Organization (WHO) criteria, Hammersmith Infant Neurological Examination Part 2 (HINE-2), and the Children’s Hospital of Philadelphia Infant Test of Neuromuscular Disorders (CHOP INTEND) scale. Participants began nusinersen treatment in infancy while still pre-symptomatic. By the time the report was published, all 25 children were past the expected age of symptom onset for SMA type I or II. At the time of analysis, all participants were alive, and none required permanent ventilation, though some needed temporary respiratory support during acute illnesses. Motor function improvements were significant: every participant achieved the ability to sit without support, 92% could walk with assistance, and 88% could walk independently. The results underscore the effectiveness of early treatment. The treatment was well tolerated, with no new safety concerns, and no participants withdrew due to treatment-related adverse events [69].

As the first disease-modifying treatment for SMA, nusinersen has been revolutionary, offering a crucial therapeutic option and marking a significant success in treating various SMA types. Its approval has transformed the ASO therapy market, spurring advancements in genetic medicine. Early initiation of nusinersen not only halts disease progression but also enables patients to achieve developmental milestones previously unattainable. It improves muscle strength, re-establishes motor neuron connections, and alleviates some respiratory issues. With high efficacy and minimal off-target effects, over 11,000 patients worldwide currently benefit from this treatment [70]. However, patients with SMA types 1 and 2 often still require assisted ventilation (non-invasive ventilation, NIV) due to ongoing respiratory challenges [71]. Despite its effectiveness, nusinersen therapy is associated with several adverse effects. Notably, in the clinical studies, 16% of patients with normal or elevated baseline platelet levels developed thrombocytopenia, and 58% of treated patients exhibited elevated urine protein levels, indicating renal toxicity, including potentially fatal glomerulonephritis. These adverse effects have led the FDA to issue warnings and recommend baseline and pre-dose monitoring of platelet counts, coagulation tests, and urine protein levels [72]. These findings underscore the importance of monitoring and optimizing ASO distribution to enhance treatment outcomes.

### 2.2. Onasemnogene Abeparvovec

#### 2.2.1. Outline

Onasemnogene Abeparvovec (ZOLGENSMA), initially known as AVXS-101, is a gene therapy aimed at treating SMA by delivering a functional *SMN1* gene to produce the vital SMN protein. Developed by AveXis (Chicago, IL, USA), the therapy became part of Novartis’ portfolio following their acquisition of AveXis in May 2018, subsequently rebranded as Novartis Gene Therapies [70]. Zolgensma received FDA approval in May 2019, followed by EMA approval in May 2020, for use in patients under two years of age with various types of SMA. This treatment is administered one time intravenously at a dose of 1.1 × 10^14^ vector genomes per kilogram of body weight, using a non-replicating, self-complementary adeno-associated virus (AAV) 9 vector to deliver the therapeutic gene [73] (Table 2).

#### 2.2.2. Discovery and Mechanism of Action

Onasemnogene abeparvovec is a gene therapy that uses an scAAV9 vector to cross the blood–brain barrier and deliver therapeutic genes [74]. Adeno-associated viruses (AAVs) are non-enveloped, single-stranded DNA viruses that require a helper virus to complete their life cycle and can transduce neurons effectively. Different AAV serotypes exhibit varying capsid properties, which influence their receptor affinity and tissue tropism [75]. The scAAV9 vectors are unique because they contain double-stranded, self-complementary DNA, which allows for faster protein synthesis after entering host cells [76]. Onasemnogene abeparvovec AAV9 vector contains human SMN1 cDNA under the control of a hybrid cytomegalovirus (CMV) enhancer/chicken-β-actin (CB) promoter. The hybrid CMV enhancer and CB promoter ensure continuous and sustained production of the SMN protein [11]. Overcoming the challenge of gene transduction into the CNS posed by the blood–brain barrier, the AAV9 vector has shown superior CNS transduction in various animal models, including mice, cats, rats, and non-human primates [74,77].

#### 2.2.3. Clinical Trials

The phase I **START** trial (NCT02122952) evaluated the safety and efficacy of onasemnogene abeparvovec. Conducted between December 2014 and August 2017, the study enrolled 15 patients with homozygous SMN1 exon 7 deletion and two copies of the *SMN2* gene (mean age: 3.4 months, range: 0.9–7.9 months). The patients were divided into two cohorts: the first cohort of three patients received a low dose of gene therapy (6.7 × 10^13^ vector genomes per kilogram [vg/kg]) over the initial four months, while the second cohort of twelve patients received a high dose (2.0 × 10^14^ vg/kg). The therapy was administered intravenously through a venous catheter inserted into a peripheral vein [78]. The primary outcome focused on determining safety by monitoring any treatment-related adverse events of grade 3 or higher. Secondary outcomes included measuring the time until death or the need for permanent ventilatory assistance (defined as at least 16 h of respiratory support per day for at least 14 days, in the absence of any acute, reversible illness or perioperative state). Additionally, motor milestones and CHOP INTEND scores were assessed as part of the exploratory outcomes. At the 18-month first data cut-off, all 15 patients in the study had a 100% survival rate at 20 months of age, with only 1 patient from the low-dose cohort requiring permanent ventilation at 29 months. All patients showed an increase in CHOP INTEND scores, with a mean increase of 24.6 points in the high-dose cohort and 16.3 points in the low-dose cohort. In the high-dose cohort, 11 out of 12 patients achieved sitting unassisted for at least 5 s, full head control, and the ability to speak; 9 of these patients could roll over, and 2 could crawl, stand, and walk independently. Seven patients in this cohort could feed orally without an enteral tube. The treatment also reduced hospitalizations per year (2.1 vs. 7.6), the median length of stay (6.7 vs. 13 days), and hospitalization duration (4.4% vs. 18.5%) compared to the natural history cohort studies, respectively [78,79]. A subgroup analysis of these 12 infants revealed that early dosing (<3 months) led to quicker achievement of motor milestones, highlighting the importance of early diagnosis and treatment [80]. During the clinical trial, one patient in the low-dose cohort experienced serious liver injury. This led to the administration of oral prednisolone (1 mg/kg/day) to the remaining 14 patients for 30 days, starting 24 h before gene vector administration. In the high-dose cohort, one patient had significantly elevated AST and ALT levels, which were managed with additional prednisolone. Two other patients had asymptomatic AST and ALT elevations that resolved without intervention [81,82].

An interventional phase III, open-label trial (**STR1VE-US**, NCT03306277) was conducted from 2018 to 2020, involving 22 SMA type 1 patients under 6 months of age who received an intravenous dose of onasemnogene. Similar phase III trials, STR1VE-EU (NCT03461289) and STR1VE-AP (NCT03837184), aimed to assess the efficacy and safety of the treatment in SMA infants of similar age and SMN2 copy number. The STR1VE-US trial has been completed, demonstrating that 91% of the enrolled patients had event-free survival at 14 months of follow-up, and 59% were able to sit without support for at least 30 s. Additionally, 81.8% of patients were free of ventilation support at 18 months old, 95% achieved a CHOP-INTEND score of 40 or higher, and 64% attained a score of 50 or higher [70,81,82].

Another clinical trial investigated the intrathecal administration of onasemnogene for patients with milder SMA (three copies of SMN2). The **STRONG** trial (NCT03381729), a phase I, open-label study, evaluated different intrathecal doses of the gene therapy (6.7 × 10^13^, 1.2 × 10^14^, or 2.4 × 10^14^ vg/kg) in 51 patients. However, a partial clinical hold was placed by the FDA due to preclinical safety concerns about dorsal root ganglia (DRG) inflammation and neuronal cell body degeneration or loss. This hold was lifted by the FDA in August 2021, and the study was completed in November 2021, demonstrating significant improvements in the HFMSE. These positive results led to the commencement of the STEER trial (NCT05089656) in February 2022 [70,82].

Finally, the phase III **SPR1N**T trial (NCT03505099) evaluated the efficacy and safety of onasemnogene abeparvovec in presymptomatic infants with spinal muscular atrophy (SMA) type 1 who have two copies of the *SMN2* gene. Conducted as a multicenter, single-arm study, the trial included 14 infants treated at six weeks or younger. Compared to a matched natural-history cohort, all infants treated with onasemnogene abeparvovec showed significant improvements. Notably, all 14 infants were able to sit independently for at least 30 s by 18 months, and 13 of them maintained a healthy body weight. None of the children required permanent ventilation or nutritional support, and no serious adverse events related to the treatment were reported. These results highlight the potential of onasemnogene abeparvovec to significantly alter the course of SMA when administered early, emphasizing the importance of universal newborn screening for SMA [83].

Onasemnogene abeparvovec has demonstrated significant benefits, including substantial improvements in motor milestone achievements and ventilator-free survival in SMA patients. These improvements offer a promising outlook for a better quality of life. However, the therapy presents several challenges. The clinical data supporting its benefits are limited to about five years, restricting the understanding of its long-term efficacy and safety. The therapy carries common adverse effects, including elevated aminotransferases and vomiting, with a significant risk of acute liver injury. This necessitates a boxed warning and rigorous liver function monitoring before and frequently after the infusion for at least three months. Baseline assessments should include liver function tests and anti-AAV9 antibodies, with systemic corticosteroids (1 mg/kg/day oral prednisolone) administered starting one day before the infusion for 30 days. Additionally, monitoring platelet counts and cardiac troponin-I levels is crucial due to transient decreases observed post-infusion. Animal studies have shown dose-dependent cardiac and hepatic toxicities, including mortality at higher doses. After the 30-day corticosteroid regimen, liver function must be reassessed. If normal, corticosteroids should be tapered off over 28 days; if abnormal, they should continue until normalization [84]. These challenges underscore the need for careful patient management and long-term monitoring to ensure the therapy’s safety and efficacy.

### 2.3. Risdiplam

#### 2.3.1. Outline

Risdiplam (EVRYSDI^®^), developed by Roche (Basel, Switzerland) (RG7916, RO7034067), is the only approved oral medication for spinal muscular atrophy (SMA). It received FDA approval in August 2020 and EMA approval in March 2021 for patients two months and older with SMA types 1, 2, and 3, or with four copies of the *SMN2* gene. In May 2022, the FDA expanded its approval to include patients of all ages, supported by promising results in treating pre-symptomatic individuals [73]. The approved dosage regimen for risdiplam is based on patient age and body weight: 0.2 mg/kg/day for patients aged 2 months to 2 years, 0.25 mg/kg/day for patients aged 2 years and older weighing less than 20 kg, and 0.5 mg/kg/day for those aged 2 years and older weighing more than 20 kg [85] (Table 2).

#### 2.3.2. Discovery and Mechanism of Action

The discovery of risdiplam was initiated through a high-throughput screening (HTS) designed to find small molecules that promote exon 7 inclusion in SMN2 pre-mRNA splicing. This process identified a promising coumarin derivative [86]. After multiple optimization efforts to address issues such as in vitro Ames flag and phototoxicity, this led to compound 2 (RG7800), a pyridopyrimidinone derivative [87]. However, the development of RG7800 was halted due to retinal toxicity observed in non-human primates [88]. Further improvements were then implemented in compound 2 to develop risdiplam, the compound 1/molecule 3 [88]. These included enhanced selectivity against off-target genes and increased on-target potency, allowing for reduced efficacious doses and a better therapeutic window. Additionally, the incorporation of a basic amine moiety with a low pKa value maintained potency while preventing hERG inhibition or phospholipidosis. The resulting compound 1, risdiplam, exhibited an excellent pharmacokinetic profile with the optimal volume of distribution and half-life, along with favorable systemic tissue distribution [89].

Mechanistically, risdiplam functions as an SMN2 splice modulator by promoting the inclusion of exon 7 in SMN2 pre-mRNA splicing, leading to the production of functional SMN protein [90]. During splicing, the U1 small nuclear ribonucleoprotein (U1 snRNP) must bind to the 5′ splice site (5′ss) at the exon–intron border. While some deviations from the 5′ss consensus sequence are tolerated, certain substitutions can weaken U1 snRNP binding, causing exon skipping [91]. SMN2 exon 7 exclusion is an example of such splice-skipping. Risdiplam-like compounds stabilize the transient double-strand RNA structure formed between the 5′ss of SMN2 exon 7 and the U1 snRNP complex. This stabilization strengthens the weak 5′ splice site of SMN2 exon 7, compensating for the sequence mismatch and increasing U1 snRNP’s binding affinity. This enhanced binding ensures proper splicing and the production of the functional SMN protein [92].

In preclinical studies, risdiplam demonstrated significant efficacy in increasing SMN protein levels and improving motor function in animal models of spinal muscular atrophy (SMA). Adult C/C allele mice treated daily with risdiplam at doses of 1, 3, or 10 mg/kg for 10 days showed elevated SMN levels, increased motor neuron counts, and enhanced neuromuscular junction (NMJ) innervation. Similarly, SMNΔ7 mice received intraperitoneal injections of risdiplam from postnatal day 3 to 9, resulting in comparable improvements [88]. In non-human primates, oral administration of risdiplam at 3 mg/kg/day for 7 days achieved good biodistribution in relevant tissues [89]. Despite initial concerns about retinal toxicity observed in monkeys, subsequent studies in pigmented and albino rats showed no retinal changes [88]. These findings supported the continued development of risdiplam, which proved more efficacious than its predecessor, compound 2, in clinical trials.

#### 2.3.3. Clinical Trials

**FIREFISH** (NCT02913482), a phase 2/3 open-label trial in SMA type I patients for risdiplam, began in December 2016 and consisted of two parts. The main focus of part 1 of the trial was to establish the safety, tolerability, pharmacokinetics, and pharmacodynamics of varying doses of risdiplam. It enrolled 21 infants aged 1 to 7 months with type 1 SMA [93]. The dosing strategy started with daily oral administration at 0.04 mg/kg, 0.08 mg/kg, or 0.2 mg/kg, depending on the age of the participant, and was adjusted to 0.2 mg/kg within a few months of starting the treatment. This was ultimately adjusted to 0.25 mg/kg when the participant reached 3 years of age. After eight months of daily treatment, 93% of participants showed significant motor improvement, with an average CHOP-INTEND score increase of 16 points and several motor milestones achieved, such as rolling, kicking, sitting with or without support, and full head control, as measured by HINE-2 [93]. A 16-month follow-up confirmed continued motor development, with 82% of the high-dose cohort reaching a CHOP-INTEND score of 40 [93,94]. None of the 90.5% (19/21) surviving infants required permanent ventilation or tracheostomies [95]. No drug-related adverse events led to withdrawal from the study after 19 months of treatment [96]. Part 2 of the trial was confirmatory, assessing the efficacy and safety of the selected dose from Part 1. After 12 months of treatment, 29% of infants could sit unsupported for at least 5 s, a milestone typically not achieved by untreated type 1 SMA patients [96]. Overall, 78% responded positively according to the HINE-2 tool, with 76% gaining head control and 61% achieving some form of sitting. Additionally, for 90% of treated infants, a significant increase in CHOP-INTEND scores was observed, with 56% achieving scores of 40 or more. By the end of the 23 months, 95% maintained the ability to swallow, with 89% able to feed orally [97,98]. Part 2 demonstrated the clinical benefit of risdiplam with a 93% survival rate. Furthermore, during the 12-month treatment course, 49% of the infants did not require hospitalization. The most common serious adverse events included respiratory issues like pneumonia and bronchiolitis, while non-life-threatening events included upper respiratory infections, pyrexia, and gastrointestinal issues such as constipation and diarrhea [98]. No retinal toxic effects as seen in cynomolgus monkeys were observed in this human study, and the treatment was generally well tolerated [93].

Another study, **SUNFISH** (NCT02908685), is a phase III, randomized, double-blind, placebo-controlled study that began in October 2016. This study enrolled children and adults aged 2–25 years with a clinical diagnosis of SMA type 2 or type 3. Similar to the FIREFISH study, the aim of this study was to test the safety, tolerability, PK, PD, and the efficacy of risdiplam and consisted of two parts. In Part 1, 51 participants were enrolled to investigate an exploratory dose over 12 weeks. Adolescents and adults aged 12 to 25 years received either 3 mg or 5 mg of risdiplam for at least 12 weeks [85]. Once the appropriate dose for Part 2 was determined, participants transitioned to the open-label study in Part 2. Part 2 involved 180 participants and was a randomized, placebo-controlled, double-blind study assessing the safety and efficacy of the dose established in Part 1. The primary endpoint was the change in MFM32 scores after 12 months of treatment. Patients treated with risdiplam showed significantly greater improvements in MFM32 scores compared to those receiving the placebo. Secondary endpoints, including the Revised Upper Limb Module (RULM) and Hammersmith Functional Motor Score-Expanded (HFMSE), also improved significantly. Patients aged 12 and older reported increased independence using the SMA Independence Scale (SMAIS). There were no major drug-related adverse events leading to withdrawal, with most events related to the underlying SMA. Common adverse events included upper respiratory infections, nasopharyngitis, pyrexia, and headache, while serious adverse events were primarily respiratory-related but not linked to risdiplam. Overall, risdiplam was well tolerated and demonstrated significant efficacy in improving motor function and independence in patients with type 2 and type 3 SMA [85,99].

**JEWELFISH** (NCT03032172), a trial investigating the effects of risdiplam in patients previously treated with other SMA therapies, began in March 2017. The trial aimed to evaluate the pharmacokinetics, pharmacodynamics, safety, and efficacy of risdiplam in non-naïve SMA patients aged 6 months to 60 years who had a clinical diagnosis of SMA type 1, 2 or 3. This multicenter, open-label study included participants who had previously received treatments such as RG7800, nusinersen, or onasemnogene abeparvovec (ZOLGENSMA) [100]. Among the 12 participants, 9 were non-ambulatory, and 3 were ambulatory, with an average MFM32 score of 48.44 at the start of the study. Over 12 months of risdiplam treatment, there was a consistent increase in SMN protein levels, averaging more than twice the baseline levels, similar to those observed in the SUNFISH Part 1 trial [101].

Finally, **RAINBOWFISH** (NCT03779334), studying risdiplam in pre-symptomatic SMA infants 0–6 weeks of age who were genetically diagnosed with SMA, began in August 2019. The primary endpoint of this study is the ability to sit unsupported for five seconds, assessed after 12 months of treatment in 10 infants with at least two SMN2 copies and a compound muscle action potential (CMAP) of ≥1.5 mV. Additional endpoints include survival, need for permanent ventilation, ability to swallow independently, CHOP-INTEND motor function score, development of SMA symptoms, and SMN protein levels in the blood. The estimated completion date of the study is early January 2029 [102].

Risdiplam has demonstrated substantial therapeutic benefits through systemic administration, as evidenced by preclinical rodent studies and clinical trials. It effectively crosses the blood–brain barrier without restricted permeability, thereby targeting the central nervous system as well as peripheral tissues, including various organs and muscles. However, administration of risdiplam is associated with some adverse effects. Commonly reported side effects include constipation, diarrhea, rash, fever, pneumonia, and vomiting. Less frequently, patients may experience urinary tract infections, joint pain, and ulcers [98]. Additionally, risdiplam may have harmful effects on the fetus when administered to pregnant women and may also compromise male fertility. Consequently, the use of risdiplam in pregnant individuals is currently advised against. Although there is a lack of comprehensive data from pregnant women undergoing risdiplam treatment, animal studies have shown risks such as embryofetal mortality, congenital malformations, and reduced fetal weight. Therefore, the prescribing information for risdiplam recommends that females undergo pregnancy testing prior to the initiation of treatment [98]. Last but not least, although risdiplam is designed to “selectively” modify the alternative splicing of the *SMN2* gene, the diversity of the spliceosome complex presents challenges. The proteins recruited by the spliceosome vary depending on the cell line, splicing event, gene type, and aging, making it difficult to ensure that a small organic molecule will exclusively target a specific splicing event. This variability raises concerns about potential off-target effects. Studies have indicated that risdiplam can induce off-target effects while targeting SMN2 exon 7 to treat SMA. These effects can impact other genes such as MBNL1, DST, TEAD1, and THOC5, leading to misidentification and non-specific recognition. These off-target effects may result in neurological disorders, muscle weakness, myalgia, muscle twitching, and paresthesia. Additionally, risdiplam’s effects on FOXM1 and MADD can disrupt cell cycles, induce micronucleus formation, and initiate apoptosis. Potential off-target impacts include thrombotic microangiopathy, urinary issues, diastolic heart failure, tachycardia, and cardiac arrest, necessitating the close monitoring of patients, particularly those with renal or cardiac conditions [103]. To enhance the quality of risdiplam treatment, further research is needed to better understand and mitigate these adverse effects. More studies should also focus on the drug’s safety profile during pregnancy and compare its efficacy and safety against other available SMA therapies.

**Table 2 genes-15-00999-t002:** Summary of FDA-approved therapies for spinal muscular atrophy (SMA) [2,35,47,50,51,52,53].

Drug/Company	Type	Mechanism of Action	Status	Route of Administration	Protocol	Targeted Population	Cost
Nusinersen (Spinraza)/Biogen	Antisense oligonucleotide	splicing modifier of SMN2 (MOE chemistry targeting SMN2 ISS-N1 that promotes exon 7 inclusion)	FDA approval—December 2016; EMA approval—May 2017	intrathecal	3 loading doses at a 14-day interval, 4th loading dose 30 days after the 3rd dose, and maintenance dose every 4 months thereafter	all ages and all types of SMA	up to USD 125,000 per dose; drug cost for the first year: USD 750,000 and then USD 375,000 annually
Onasemnogene abeparvovec-xioi (Zolgensma)/Novartis	Gene therapy	replacement of SMN1 gene (scAAV9-SMN under the control of a CBA promoter)	FDA approval—May 2019; EMA approval—conditional approval on May 2020	intravenous	Single dose	FDA: SMA patients less than 2 years of age with bi-allelic mutations in the *SMN1* gene. EMA: patients with 5q SMA with a bi-allelic mutation in the *SMN1* gene and a clinical diagnosis of SMA type 1, or patients with 5q SMA with a bi-allelic mutation in the *SMN1* gene and up to 3 copies of the *SMN2* gene	USD 2,125,000 (single injection)
Risdiplam (Evrysdi)/Roche	Small molecule	splicing modifier of SMN2	FDA approval—August 2020; EMA approval March 2021	orally	Once daily	patients 2 months of age and older	up to USD 340,000 a year (cheaper in younger patients as dosing depends on weight)

## 3. Broader Therapeutic Strategies: SMN-Dependent and -Independent Approaches

Research continues to develop drugs for SMA by targeting and increasing SMN expression via splice modulation, leading to the development of additional drugs targeting SMN2.

**Branaplam:** Branaplam (also known as LMI070), similar to risdiplam, is an orally available small molecule developed by Novartis as a potential therapy for spinal muscular atrophy (SMA). It was identified through a high-throughput screen designed to find molecules that promote the inclusion of exon 7 in SMN2 pre-mRNA, stabilizing it with splicing factor complexes. In SMA mice, daily administration of Branaplam resulted in a dose-dependent increase in exon 7 inclusion and SMN protein expression, improving body weight and lifespan. The first human phase I/II trial (NCT02268552) for Branaplam began in 2015, targeting SMA patients under six months old with two copies of SMN2. This trial aimed to evaluate the safety, tolerability, and early effectiveness of Branaplam [11]. However, recruitment was halted in 2016 due to adverse events observed in animal studies while the trial was underway, which included damage to nerves, the spinal cord, testes, and kidney blood vessels. Enrollment resumed in late 2017 with modifications to the study design, including the allowance of both feeding tubes and oral administration and the addition of nerve tests as safety measures. Recruitment was completed in May 2019, encompassing 13 infants in part 1 and 25 infants in part 2. Later that year, the company announced positive progress, with some infants receiving therapy for more than four years, though full results were not disclosed, leaving the complete safety profile unknown (branaplam (LMI070) (https://smanewstoday.com/)). Additionally, clinical deterioration was observed after reducing the subsequent target dose to one-tenth. Due to rapid advancements in SMA treatments, Novartis discontinued Branaplam’s development for SMA in mid-2021 [70]. However, Branaplam has shown potential in reducing huntingtin mRNA, the mutated protein in Huntington’s disease, earning the FDA Orphan Drug Designation for Huntington’s, with a phase IIb trial planned for 2021 [11].

**Histone deacetylase inhibitors (HDAC):** Histone deacetylase (HDAC) inhibitors have been extensively investigated for their role in treating spinal muscular atrophy (SMA) due to their ability to activate SMN2 transcription by inhibiting deacetylation of chromatin histones, thereby promoting gene expression [104,105,106,107]. Notably, several HDAC inhibitors, including valproic acid, trichostatin A, and sodium phenylbutyrate (Figure 2), have demonstrated promising results in increasing SMN levels both in vitro and in vivo [108]. Valproic acid, a classic class I HDAC inhibitor, has shown beneficial effects in SMA mouse models and patient fibroblasts, leading to its rapid progression to clinical trials [109]. However, a systematic review of these trials indicated an improvement in motor function but little effect on survival [110]. Similarly, another class I HDAC inhibitor, phenylbutyrate, was shown to increase SMN protein levels and the number of Gems in SMA fibroblast culture [107]; however, a randomized, placebo-controlled trial showed no significant improvement in motor function, leading to the premature termination of its clinical trial (NCT00439569) [111]. Other small molecules with HDAC inhibitor properties, such as suberoylanilide hydroxamic acid (SAHA/Vorinostat) [112], trichostatin A [113], and resveratrol [114], also demonstrated success in laboratory models of SMA but have not advanced to clinical trials. While HDAC inhibitors alone do not provide therapeutic benefits equivalent to SMN replacement, they may offer additional neuroprotective support when used in combination with other SMN-targeting therapies [1,110]. This concept is supported by recent research demonstrating the advantages of combining the HDAC inhibitor LBH589 (panobinostat) with low doses of Spinraza-like antisense oligonucleotides (ASOs), suggesting a potential synergistic approach for enhancing SMA treatment efficacy [115]. A study by Marasco et al. demonstrated that while nusinersen-like ASOs promote exon 7 inclusion in the *SMN2* gene, they also induce a silencing mark (H3K9me2) that inhibits this process. The histone deacetylase inhibitor VPA counteracts this inhibition by promoting transcriptional elongation and cooperates with ASO, thereby enhancing its effectiveness. Using HDAC inhibitors like VPA in conjunction with ASOs increases SMN protein levels, which improves growth, survival, and neuromuscular function in SMA treatment [116].

**Neuromuscular junction-targeting therapies:** SMA is associated with an impairment of neuromuscular junction dysfunction (NMJ) development, maturation, and function, contributing to muscle weakness and fatigue. Targeting NMJ pathology presents a potential complementary therapy for SMA. Therapeutic strategies targeting NMJ under development include Salbutamol, Amifampridine, NT-1654, and Pyridostigmine [11,70] (Figure 2).

Salbutamol/albuterol, β-adrenoreceptors agonists, primarily act on Beta2-adrenoreceptors in the smooth muscle of blood vessels, lungs, and intestines. They have shown promise in improving muscle strength by increasing full-length SMN mRNA, SMN protein, and Gem numbers by promoting exon 7 inclusion in SMN2 in vitro and possibly stabilizing acetylcholine receptor clusters at the NMJ [117,118]. Although small clinical studies suggest benefits in maintaining motor function or improving respiratory function in SMA type 2 patients, large-scale, placebo-controlled studies are lacking [70].

Amifampridine, a voltage-dependent K+ channel blocker, enhances neuromuscular transmission and muscle function. Approved for Lambert–Eaton myasthenic syndrome [11], it was also tested in a randomized, double-blind, placebo-controlled crossover study (NCT03781479) for SMA involving 13 type 3 SMA patients who could walk unaided. The primary outcome was the HFMSE score change, with secondary outcomes including timed tests and quality-of-life assessments. The study indicated that patients receiving amifampridine showed a mean HFMSE improvement of 0.792 compared to the placebo group, with no serious adverse events reported. This suggests that the NMJ could be a potential SMN-independent therapeutic target, highlighting the need for larger studies to confirm amifampridine’s role as an adjunctive therapy in SMA treatment [119].

Pyridostigmine, an acetylcholinesterase inhibitor, slows the degradation of acetylcholine within the synaptic cleft, enhancing cholinergic transmission efficiency. It is used to improve muscle strength in myasthenia gravis, an autoimmune disorder that also causes muscle weakness. The SPACE trial (NCT02941328) is a phase 2, monocenter, double-blind, placebo-controlled cross-over trial to assess the efficacy of the pyridostigmine in SMA type II-IV involving 37 participants aged 12 and older. The trial assessed motor fatigue and function using the repeated nine-hole peg test (R9HPT) and the motor function measure (MFM). Results showed no significant difference in R9HPT scores between treatments, and while MFM scores were slightly better with pyridostigmine (average 42.4% vs. 41.6%), the difference was not statistically significant. However, 74.4% of pyridostigmine-treated patients reported less fatigue compared to 29.7% on placebo, indicating a significant reduction in fatigue [120].

Additionally, subcutaneous administration of NT-1654, the active portion of agrin, has been shown to delay disease progression in SMA mouse models [121]. The agrin/MuSK signaling pathway, crucial for NMJ formation and maturation, is dysregulated in SMA. Overexpressing agrin or administering agrin-like molecules or downstream mediators like DOK7 can enhance NMJ structure and reduce disease severity in SMA [122,123].

**Neuroprotective and muscle-enhancing therapies:** Recent studies have shown that selectively depleting SMN in the skeletal muscle of mice leads to muscle and neuromuscular junction (NMJ) pathology. It is hypothesized that improving muscle pathology may help preserve proprioceptive synapses on motor neurons, which are typically lost in SMA. Consequently, targeting muscle is seen as a promising therapeutic approach through various strategies (Figure 2).

Myostatin inhibitors: Myostatin acts as a negative regulator of muscle growth. Inhibition of the myostatin signaling pathway has shown promising results, especially in less severe models of SMA or in addition to SMN-restoring therapies. Myostatin-deficient animals and humans demonstrate significantly increased musculature [123,124].

Apitegromab (SRK-015) is an investigational fully human monoclonal antibody which inhibits the activation of myostatin, thereby preserving muscle mass. In SMA mice, apitegromab improved muscle mass and function [125]. The phase 2 TOPAZ clinical trial evaluated the safety and efficacy of apitegromab in nonambulatory patients with SMA types 2 and 3, including 58 participants aged 2 to 21 years. The trial demonstrated sustained improvements in motor function over 24 months, particularly in younger patients given higher doses. The trial’s extension phase of up to three years confirmed these positive outcomes, with patients showing continued improvement in motor abilities and reduced fatigue. Apitegromab was well tolerated, with minor adverse effects such as headache, upper respiratory tract infections, pyrexia, nasopharyngitis, cough, and vomiting [126].

Follistatin is an endogenous antagonist of myostatin; the over-expression of recombinant follistatin in SMA mouse muscle leads to increased skeletal muscle mass and survival [127].

A hybrid activin II receptor (ACTIIR) ligand trap, BIIB110 (ALG 801), inhibits activin receptor type IIB (ActRIIB) ligands, promoting muscle growth, which in turn enhances muscle mass and function. It is currently in phase 1 of clinical development [53].

Fast skeletal muscle troponin activators: Reldesemtiv (formally CK-2127107) enhances muscle contractility by prolonging calcium binding to the troponin complex in fast skeletal muscles, reducing the energetic cost of muscle contraction [128]. In a phase 2 trial (NCT02644668) for SMA type II–IV patients, participants received either 150 mg or 450 mg twice daily for 8 weeks. The higher-dose group showed better performance in the 6-minute walk test (6MWT) and the Timed Up and Go (TUG) test, as well as increased maximal expiratory pressure (MEP). Some patients reported serious adverse events such as elevated blood creatine phosphokinase and aspartate aminotransferase levels, along with gastrointestinal infections [129]. Further studies are required to establish its efficacy, including in type 1 SMA infants.

Neuroprotection: In SMA, neurons and muscles—the primary tissues affected—have high energy demands, making energy pathways potentially neuroprotective and therapeutic [130]. Olesoxime (TRO19622), a member of the trophos cholesterol-oxime compound family, functions as a mitochondrial pore modulator with neuroprotective properties. Pre-clinical studies indicated its potential to enhance neuron function and survival [131]. However, a phase 2 placebo-controlled trial in patients with types 2 and 3 SMA revealed only a modest but consistent benefit compared to the placebo. A subsequent 18-month follow-up study (OLEOS, NCT02628743) failed to demonstrate significant clinical benefits, leading the pharmaceutical company to halt its development for SMA in June 2018 [132]. Patients from this study were subsequently enrolled in the JEWELFISH (RO7034067) clinical trial with risdiplam [70].

Other potentially neuroprotective agents, such as riluzole and gabapentin, have been explored for their effects on treating SMA by targeting excitotoxicity. Riluzole, known for its modest effects in ALS, was tested in a small preliminary phase I trial involving seven participants, which suggested some potential benefits [133]. Subsequently, a phase II/III multicenter, randomized, double-blind study was conducted to assess riluzole’s efficacy and safety in young adults with types 2 and 3 SMA (NCT00774423). However, the majority of the results were not encouraging, and the studies failed to demonstrate significant efficacy [108,134]. Similarly, gabapentin was tested in type 2/3 SMA patients due to its effects in ALS. While one study showed some improvement in motor function, another did not demonstrate any significant benefit [135,136].

**Targeting cell death mechanism:** In SMA, neurodegeneration is linked to disruptions in core pathways of cell homeostasis, including autophagy, ubiquitin homeostasis, and apoptotic pathways [137,138]. Therapeutic strategies targeting molecules involved in these pathways are under investigation. Celecoxib (Figure 2), a selective cyclooxygenase-2 (COX-2) inhibitor that can cross the blood-brain barrier, has shown potential for increasing SMN protein levels through the activation of the p38 pathway in SMA cell and rodent models [139]. However, despite these promising preclinical findings, a phase II clinical trial of celecoxib for SMA was prematurely terminated, and the results have not yet been published (NCT02876094) [11].

**Targeting cytoskeleton:** In SMA, the upregulation of the RhoA/Rho kinase (ROCK) pathway disrupts actin dynamics, which disrupts cytoskeleton structure and hinders neuronal growth and regeneration [140]. Therefore, targeting this pathway can be a possible therapeutic option. Y-27632 [141] and the FDA-approved fasudil [142], pharmacological inhibitors of the ROCK pathway (Figure 2), have demonstrated improvements in survival, neuromuscular junction (NMJ) maturation, and muscle development in SMA mouse models.

## 4. Challenges in the Treatment Era: Are We There Yet in the Battle against SMA?

The therapeutic landscape for spinal muscular atrophy (SMA) has seen remarkable advancements with significant progress in extending patient lives and improving motor function. Despite the success of FDA-approved treatments like nusinersen, onasemnogene abeparvovec, and risdiplam in improving patient outcomes, these therapies come with several challenges.

Nusinersen, for instance, must be injected directly into the cerebrospinal fluid (CSF) intrathecally to ensure it reaches the CNS to treat lower motor neurons, although this method is often associated with thrombocytopenia and injection-site adverse events [55,143,144]. Adult SMA patients face additional challenges such as scoliosis, spine or thorax deformities, and prior spinal surgeries, which complicate the safe administration of nusinersen [145]. Furthermore, intrathecal administration limits nusinersen’s efficacy to the central nervous system, failing to address SMA’s multi-organ impact effectively [146]. Recent studies in animal models and humans indicate that SMA affects multiple organs, including the heart, peripheral nervous system, skeletal muscle, liver, and vasculature [5,147]. For example, muscle-specific SMN loss in SMA mouse models results in compromised motor performance, premature death, muscle fiber defects, and neuromuscular junction abnormalities. In human fetuses with SMA, SMN deficiency causes delayed growth and maturation of myotubules [148]. Untreated severe SMA patients with only one copy of SMN2 have shown thrombotic occlusions of small blood vessels, leading to digital necrosis, suggesting that severe SMN deficiency may also manifest as a vascular disease [149]. The increasing evidence of multi-organ involvement in SMA suggests that ASO therapies targeting motor neurons alone may be insufficient for the long-term management of SMA pathology. Additionally, a recent study [150] investigating nusinersen’s biodistribution in post-mortem infant spinal cord samples found variability in ASO distribution, with notably lower concentrations in the cranial portion of the spinal cord and the brain, suggesting uneven distribution within the CNS and potentially affecting treatment efficacy. Despite positive outcomes from pre-symptomatic intrathecal treatment in the NURTURE trial, the long-term effects and efficacy of nusinersen remain uncertain [67,68]. Furthermore, gaps remain in understanding whether nusinersen helps achieve significant milestones, regain function, or avoid serious side effects and ventilation support. More research and long-term data are needed to justify its use in specific SMA patient subgroups.

Onasemnogene abeparvovec (Zolgensma) is a one-time gene therapy that offers significant survival benefits for SMA patients. However, it raises concerns about liver toxicity and reduced efficacy over time [151,152]. This reduction happens because cell division dilutes the episomal SMN cDNA within the target cell, limiting expression to non-dividing cells like neurons [11]. The body’s immune response to the viral vector complicates repeated treatments, while systemic delivery and maintenance of the treatment are crucial for long-term benefits due to the involvement of peripheral organs in the disease. Onasemnogene has to be administered within the first two years of life, making newborn screening for SMA crucial for early diagnosis and timely treatment, given the limited window for effective intervention. Serious side effects, such as liver damage, thrombotic microangiopathy, thrombocytopenia, and cardiac toxicity have also been observed for onasemnogene and therefore require careful patient monitoring. Furthermore, recent studies have highlighted the toxic gain of function of SMN, especially in the sensorimotor circuit, including the loss of proprioceptive neurons. Similar results were also found when intrathecal injection was explored as an alternative delivery method (NCT03381729). Pre-clinical trials involving intrathecal injection have shown concerns about neurotoxicity, including ataxia, loss of proprioceptive synapses, neuroinflammation, and neurodegeneration [153,154,155]. These issues highlight the need for more long-term data to fully understand the therapy’s effectiveness and safety.

Risdiplam, an oral medication, provides a more accessible treatment option, especially for older patients, after receiving its approval in 2020 for all SMA types in patients 2 years of age. Although it overcomes the necessity for invasive intrathecal injections, its long-term effects are still unknown. Potential risks include male fertility impairment, growth issues, retinal toxicity, and off-target effects, although these have not yet been observed in humans [98,156,157]. The drug is contraindicated for patients with hepatic abnormalities, which could limit its use in many older SMA patients who often present with liver issues.

Apart from these, the extremely high cost of these treatments poses a significant barrier to widespread accessibility. While early treatment is critical for optimal outcomes [158], there remains a subset of patients who do not respond to these therapies [67]. The variability in response can be attributed to factors beyond the SMN1 and SMN2 genes, including other genetic factors, environmental influences, and disparities in medical care quality and accessibility.

## 5. Future Directions: Advancing the Battle Against SMA

The field of spinal muscular atrophy treatment has significantly evolved with the introduction of nusinersen, onasemnogene abeparvovec, and risdiplam, which have become the standard of care. Despite their ground-breaking nature, it has become clear through long-term follow-ups that *SMN2* gene modulation or *SMN1* gene replacement alone does not constitute a cure. Future directions in SMA treatment involve incorporating SMN-independent therapies alongside SMN-dependent ones to provide broader benefits, especially for patients diagnosed later or with milder forms of SMA. A cross-disease approach, repurposing drugs with established safety profiles, could also expedite the development of effective therapies. Additionally, combining currently approved therapies has garnered interest due to their different approaches to treating SMA. For example, Lee et al. provided insights on patients first treated with nusinersen followed by onasemnogene [159], while another small study involving five patients treated with both therapies showed phenotype improvements but highlighted potential liver toxicity risks [160]. More research is needed to determine the therapeutic benefits and cost-benefit analysis of combination treatments.

In addition to drug-based therapies, a multidisciplinary approach is essential for effective SMA treatment, incorporating physiotherapy, rehabilitation, respiratory management, orthopedic care, and nutritional support [161]. Further research is needed to determine the most effective combination of these treatments.

Recent evidence strongly supports that early intervention provides greater clinical benefits than treating symptomatic patients, with a high potential to achieve age-appropriate motor milestones [158]. Therefore, prenatal and neonatal genetic screening is crucial for early diagnosis and treatment. Prenatal genetic counselors can play a key role in this process. Improved education and interdisciplinary communication are needed to ensure that counselors can provide the necessary information, empowering patients to make informed decisions about their pregnancies and potential treatments. Additionally, good training of medical personnel in recognizing and diagnosing SMA is essential due to the complexity and variability of neuromuscular diseases. Proper training ensures early detection, which is crucial for effective treatment and improving patient outcomes. Moreover, obtaining more than one medical consultation is important because it reduces the risk of misdiagnosis and ensures a comprehensive evaluation. Multiple consultations can help confirm the diagnosis more quickly, which is vital given that delays in diagnosis can exceed the average life expectancy of SMA patients, thereby affecting their treatment and quality of life. Highlighting the significance of newborn screening for SMA, scientists completed an observational newborn screening study in Belgium titled “Sun May Arise on SMA” [162]. Similarly, Prof. Servais launched a newborn screening study in Oxford, UK, in March 2022, expected to conclude in August 2025 [163].

Finally, research to improve existing treatment approaches for SMA is essential to offer more effective, safer, and more accessible options. One promising area of research focuses on enhancing the efficacy and minimizing the toxicity of antisense oligonucleotides (ASOs), as nusinersen’s success highlights the promise and efficiency of ASO-based therapies, yet there is room for enhancement. A notable advancement is the development of phosphorodiamidate morpholino oligomers (PMOs), different chemistry of ASO, which are neutrally charged due to their phosphorodiamidate linkage and offer enhanced stability and minimal toxicity [164]. These properties have made PMOs effective therapeutic options, as evidenced by the approval of four PMO-based ASOs for Duchenne Muscular Dystrophy (DMD) [165]. In the context of SMA, studies on mouse models have demonstrated that PMOs can significantly increase SMN2 levels and improve survival rates through various delivery routes, including intrathecal and intravenous injections, although the superiority of these delivery routes remains debated. Most notably, PMOs have shown lower toxicity, prolonged survival, and, most importantly, effective SMN restoration in CNS tissues. Additionally, PMOs’ neutral charge allows them to be conjugated with cell-penetrating peptides (CPPs) for improved delivery and uptake in target tissues [166,167,168]. CPPs, short sequences that facilitate the uptake of cargo into target cells, were discovered in the late 1980s [165]. In the context of SMA, CPPs such as Pip6a, RXR peptides, and DG9 have shown promising results by enhancing cellular uptake and crossing the blood-brain barrier, leading to better restoration of SMN levels, prolonged survival, and improved phenotypes in mouse models [169,170,171]. Overall, it is crucial to continue researching ways to enhance current treatment approaches while exploring new options to ensure the best possible outcomes for SMA patients.

## 6. Conclusions

The field of spinal muscular atrophy (SMA) treatment is witnessing unprecedented advancements, bringing life-changing options to patients and their families. The approval of three SMA therapies—nusinersen, onasemnogene abeparvovec, and risdiplam—has opened new possibilities and transformed the standard of care. However, several challenges persist. Addressing these challenges in the coming years will be critical for the future development of SMA therapies. Continued research is essential, particularly for chronic forms of SMA and to establish optimal treatment regimens. Monitoring the evolution of new phenotypes and complications that have not been previously observed is also necessary. A combination of therapies and a multidisciplinary approach may play a significant role in future developments, though testing these approaches will be complex given the evolving phenotypes. Moreover, as more therapies become available, it will be crucial to consider the ethical and practical implications for all stakeholders, including the healthcare system, industry, patients, and caregivers. Overall, advancing SMA treatment will require a holistic and collaborative effort to ensure the best outcomes for patients.

## Figures and Tables

**Figure 1 genes-15-00999-f001:**
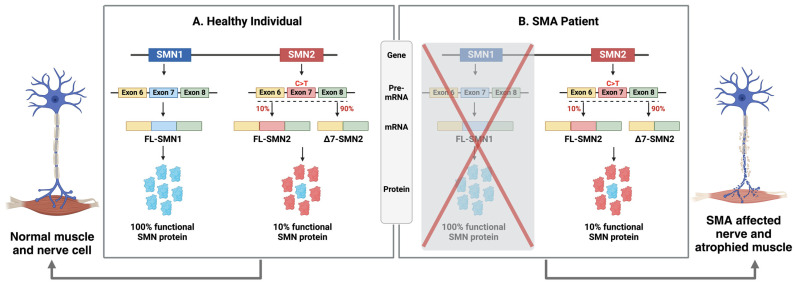
Genetic basis of SMA. (**A**) Healthy individuals produce 100% functional SMN protein from their *SMN1* genes and only 10% from each of their *SMN2* genes. (**B**) Patients with SMA do not have functional SMN1 and rely on the 10% SMN protein produced by SMN2 due to a C to T transition mutation in exon 7. Most of the SMN protein encoded by SMN2 is rapidly degraded (created with BioRender.com).

**Figure 2 genes-15-00999-f002:**
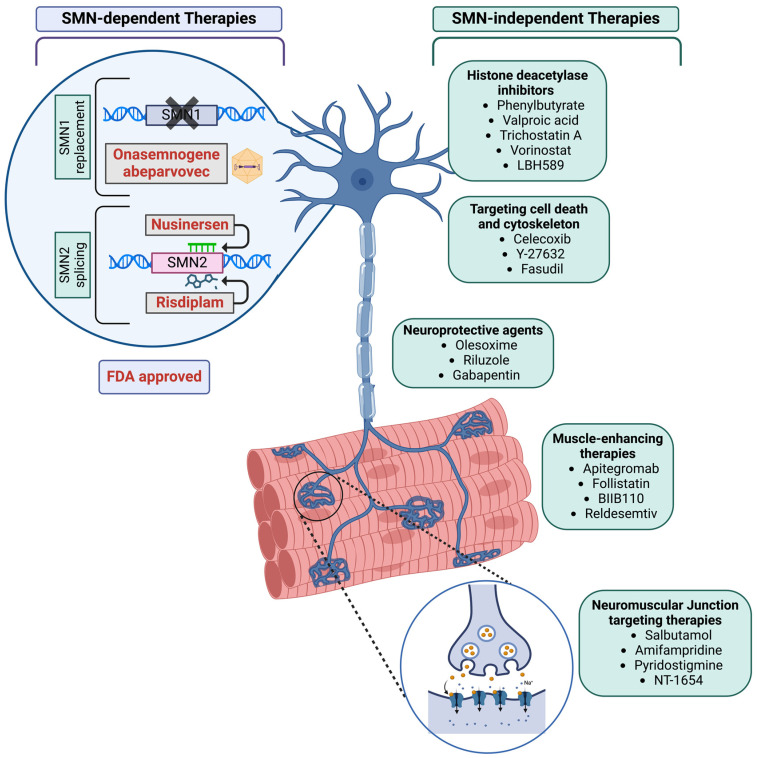
Schematic diagram of SMN-dependent (FDA-approved) and SMN-independent therapies (in development) (created with BioRender.com).

**Table 1 genes-15-00999-t001:** Classification of spinal muscular atrophy (SMA) [2,35,47,50,51,52,53].

SMA Type	Age of Onset	Life Expectancy	SMN2 Copy Number	Clinical Manifestation	Alternative Name	Estimated SMA Portion
0	Prenatal	<1month	1	Require assisted respiration at birth; fetus displays reduced movement	-	Unclear, maybe <1%
1	0–6 months	<2 years	2	Unable to sit independently, respiratory and feeding support required	Werdnig–Hoffman disease	~60%
2	<18 months	>2 years	3, 4	Ability to independently sit, inability to walk, respiratory (often non-invasive) and feeding support required	Dubowitz disease	~27%
3a	18 months–3 years	Adult	3, 4	Full ambulation, but slowly progressive muscle atrophy and weakness.	Kugelberg–Welander disease	~12%
3b	>3 years	Adult	4			
4	>21 years	Adult	≥4	Usually preserved walking ability	Adult-onset SMA	~1%

## Data Availability

No new data were created or analyzed in this study. Data sharing is not applicable to this article.

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
