# Peer review of "Recent Progress in Gene-Targeting Therapies for Spinal Muscular Atrophy: Promises and Challenges"

_genes, 2024, doi:10.3390/genes15080999_

Round 1

Reviewer 1 Report

Comments and Suggestions for Authors

The review by Haque et al. is a very well written, carefully detailed review and is an excellent compendium of the latest publications related to new ways of treating SMA. I think it is very important to accept this review because it will help the scientific community and the families affected by the disease.

Minor Comments

·       The part about Nusinersen is impeccable, there is not much to add. Optionally, you could add a section on the most recent studies (2021 onwards) or explain how this influenced the current market for ASO therapies. I also think it is worth noting that this type of therapy allows for the suspension of treatment in the face of adversity and that it showed high efficiency with low off targets.

·       In the line 388 where states than Onasemnogene is better than Nusinersen, regarding Onasemnogene being more efficient in administration compared to Spinraza, it is a relative comparison. First of all Spinraza is not a gene therapy, and therefore gives the option to the patient or the physician to interrupt the therapy in case of counterproductive effects or in case of extra treatment. This modularity of administration of Nusinersen, although unfortunately intrathecal, is much more harmless and reversible than Onasemnogene's gene therapy. Considering that during the clinical trials of Onasemnogene discrepancies arose regarding data manipulation and delayed reporting, and the lack of guaranteed improvement with respect to being a single-dose treatment, it does not make it one of the most optimal forms of therapy (whereas there is also no way to ensure that the virus is not integrated into the patient's genome).

·       Regarding the Risdiplam part, I think it is very well detailed. But I think it would be important to point out that although technically it is a small organic molecule that "selectively" modifies the alternative splicing of the SMN2 gene. But the spliceosome complex is very diverse, and the proteins it recruits are variable with respect to cell line, splicing event, gene type or aging, so I think it is difficult to say that a small organic molecule will only affect a particular splicing event, which raises the question of the possible offtargerts it can generate. However, it is also important to note that it is the only treatment that has a non-invasive form of administration for the patient, which makes it possible to be used as a complementary therapy.

·       In the section on HDAC inhibitors, in citation 116, I think it can be complemented with another recent work where they show that an ASO can generate chromatin methylation (and subsequent compactation and RNAPII roadblocks), explaining why the co-administration of a HDACi could benefit the Nusinersen theraphy.

·       I really like Figure 2 and the section on compounds with neuroprotective and neuromuscular functions

·       In the future directions part, I think it is very important to highlight the good training of medical personnel when it comes to recognizing and diagnosing neuromuscular diseases. Also the importance of obtaining more than one medical consultation, since many times the average time for a correct diagnosis of patients with SMA is longer than their average life.

·       I like what lines 828 says. It seems to me that the importance of government policies when carrying out prenatal screenings and offering treatments deserves to be mentioned, since we are in a transition where today there are multiple therapies for a disease that until now little had no cure. Diagnosing and treating a patient with SMA at the right time can change the life of that patient and their family.

I think the review is very carefully written, but I think you could add a part where you remember that research in basic science, and basic molecular biology mechanisms, allowed Krainer's group to develop the first effective therapy for SMA. It is important to highlight this, since the rest of the proposed therapies were developed under different ways of doing science, without being guided by hypotheses based on observations, and carried out simply by the high output of screenings as in the case of Risdiplam. I think it is vitally important to highlight that basic science is the engine of scientific-technological advances.

Author Response

RESPONSE TO REVIEWERS:

We appreciate all the valuable comments from the Reviewers. We have provided additional data and have revised the manuscript according to the Reviewers’ comments and suggestions. We believe that the manuscript has been further improved. Our point-by-point response to the comments and suggestions, and the corresponding revisions and modifications in the manuscript are described below.

RESPONSE TO REVIEWER #1

Comment 1: The part about Nusinersen is impeccable, there is not much to add. Optionally, you could add a section on the most recent studies (2021 onwards) or explain how this influenced the current market for ASO therapies. I also think it is worth noting that this type of therapy allows for the suspension of treatment in the face of adversity and that it showed high efficiency with low off targets.
Response 1: We deeply appreciate the valuable suggestion provided by the reviewer. In response to this recommendation, we have made added the required information: “As the first disease-modifying treatment for SMA.............currently benefit from this treatment” in page-8.

Comment 2: In the line 388 where states than Onasemnogene is better than Nusinersen, regarding Onasemnogene being more efficient in administration compared to Spinraza, it is a relative comparison. First of all Spinraza is not a gene therapy, and therefore gives the option to the patient or the physician to interrupt the therapy in case of counterproductive effects or in case of extra treatment. This modularity of administration of Nusinersen, although unfortunately intrathecal, is much more harmless and reversible than Onasemnogene's gene therapy. Considering that during the clinical trials of Onasemnogene discrepancies arose regarding data manipulation and delayed reporting, and the lack of guaranteed improvement with respect to being a single-dose treatment, it does not make it one of the most optimal forms of therapy (whereas there is also no way to ensure that the virus is not integrated into the patient's genome).
Response 2: We express our gratitude for the invaluable feedback provided. In accordance with the reviewer's suggestion, we have removed the sentence in page- 10.

Comment 3: Regarding the Risdiplam part, I think it is very well detailed. But I think it would be important to point out that although technically it is a small organic molecule that "selectively" modifies the alternative splicing of the SMN2 gene. But the spliceosome complex is very diverse, and the proteins it recruits are variable with respect to cell line, splicing event, gene type or aging, so I think it is difficult to say that a small organic molecule will only affect a particular splicing event, which raises the question of the possible offtargerts it can generate. However, it is also important to note that it is the only treatment that has a non-invasive form of administration for the patient, which makes it possible to be used as a complementary therapy.

Response 3: We deeply appreciate the valuable suggestion provided by the reviewer. In response to this recommendation, we have provided a concise description of the off-target effect of risdiplam in page 13 of our manuscript, which reads: “Last but not the least, although risdiplam is designed to "selectively" .................particularly those with renal or cardiac conditions”.

Comment 4: In the section on HDAC inhibitors, in citation 116, I think it can be complemented with another recent work where they show that an ASO can generate chromatin methylation (and subsequent compactation and RNAPII roadblocks), explaining why the co-administration of a HDACi could benefit the Nusinersen therapy.

Response 4: We deeply appreciate the valuable suggestion provided by the reviewer. In response to this recommendation, we have included the recent work on HDACi in page- 15, which specifically reads, “A study by Marasco et al. demonstrated that while nusinersen-like ASOs promote exon 7 inclusion in the SMN2 gene, they also induce a silencing mark (H3K9me2) that inhibits this process. The histone deacetylase inhibitor VPA counteracts this inhibition by promoting transcriptional elongation and cooperate with ASO, thereby enhancing its effectiveness. Using HDAC inhibitors like VPA in conjunction with ASOs increases SMN protein levels, which improves growth, survival, and neuromuscular function in SMA treatment.”

Comment 5: I really like Figure 2 and the section on compounds with neuroprotective and neuromuscular functions.

Response 5: We are truly grateful for your appreciation of figure 2.

Comment 6: In the future directions part, I think it is very important to highlight the good training of medical personnel when it comes to recognizing and diagnosing neuromuscular diseases. Also the importance of obtaining more than one medical consultation, since many times the average time for a correct diagnosis of patients with SMA is longer than their average life.

Response 6: Thank you for your valuable suggestion regarding the inclusion of importance of the good training of medical personnel in the future direction of the manuscript. We have provided the required information in page- 21 which reads: " Additionally, good training of medical personnel in recognizing and diagnosing SMA is essential due to the complexity and variability of neuromuscular diseases. Proper training ensures early detection, which is crucial for effective treatment and im-proving patient outcomes. Moreover, obtaining more than one medical consultation is important because it reduces the risk of misdiagnosis and ensures a comprehensive evaluation. Multiple consultations can help confirm the diagnosis more quickly, which is vital given that delays in diagnosis can exceed the average life expectancy of SMA pa-tients, thereby affecting their treatment and quality of life."

Comment 7: I like what lines 828 says. It seems to me that the importance of government policies when carrying out prenatal screenings and offering treatments deserves to be mentioned, since we are in a transition where today there are multiple therapies for a disease that until now little had no cure. Diagnosing and treating a patient with SMA at the right time can change the life of that patient and their family.

Response 7: We are grateful for your appreciation and sincerely appreciate your valuable comment regarding the importance of prenatal screening. In page 21 we have added the information, which reads: “Prenatal genetic counselors can play a key role in this process. Improved education and interdisciplinary communication are needed to ensure that counselors can provide the necessary information, empowering patients to make informed decisions about their pregnancies and potential treatments.”

Comment 8: I think the review is very carefully written, but I think you could add a part where you remember that research in basic science, and basic molecular biology mechanisms, allowed Krainer's group to develop the first effective therapy for SMA. It is important to highlight this, since the rest of the proposed therapies were developed under different ways of doing science, without being guided by hypotheses based on observations, and carried out simply by the high output of screenings as in the case of Risdiplam. I think it is vitally important to highlight that basic science is the engine of scientific-technological advances.

Response 8: We deeply appreciate the valuable suggestion provided by the reviewer. In response to this recommendation, we have included the required information: “A pivotal moment…… the successful development of treatments like nusinersen on page 5-6.

Reviewer 2 Report

Comments and Suggestions for Authors

The paper "Recent Progress in Gene-Targeting Therapies for Spinal Muscular Atrophy: Promises and Challenges" provides a thorough overview of the current state of disease modifying therapies for spinal muscular atrophy (SMA). The authors comprehensively review both approved treatments and those still under investigation, detailing their mechanisms of action, clinical study results, challenges, and potential avenues for improvement through combination therapy.

Some minor corrections should be made in the document (for this reason, I attach a .pdf file with comments). Generally, authors should use the word “absence” instead of “deletion”, since, as authors mentioned in one part, an absence can also be due to gene conversion. Also, authors introduce the same abbreviation several times and this should be corrected.  Notwithstanding these minor corrections, the overall quality and usefulness of the paper remain unimpaired.

This review is an indispensable resource for researchers, clinicians, and patients seeking a comprehensive understanding of genetically designed therapies for SMA. The authors' comprehensive overview provides a valuable roadmap for future research directions and highlights the potential for combination therapies to bring us closer to a cure for this devastating disease.

Author Response

RESPONSE TO REVIEWER #2

Comment 1: Some minor corrections should be made in the document (for this reason, I attach a .pdf file with comments). Generally, authors should use the word “absence” instead of “deletion”, since, as authors mentioned in one part, an absence can also be due to gene conversion. Also, authors introduce the same abbreviation several times and this should be corrected.  Notwithstanding these minor corrections, the overall quality and usefulness of the paper remain unimpaired.

Response 1: The concern has been addressed and all the required changes have been made in the revised manuscript.